# Predictors of Impaired Treatment Outcomes in COVID-19: A Single-Center Observational Study from Serbia

**DOI:** 10.3390/diagnostics15131685

**Published:** 2025-07-02

**Authors:** Tatjana Adzic-Vukicevic, Milan Racic, Nikolina Tovarisic-Racic, Marija Laban-Lazovic, Sead Dalifi, Jovana Radmilovic

**Affiliations:** 1School of Medicine, University of Belgrade, 11000 Belgrade, Serbia; adzic_tatjana@yahoo.com; 2Clinic of Pulmonology, University Clinical Center of Serbia, 11000 Belgrade, Serbia; marija.laban.lazovic@gmail.com (M.L.-L.); seaddalifi@yahoo.com (S.D.); 3Service for Pulmonary Disease and Tuberculosis, Health Center, 81400 Niksic, Montenegro; race.mr211@yahoo.com; 4Service for Microbiology, Health Center, 81400 Niksic, Montenegro; nikolinat99@gmail.com; 5Center for Radiology and Magnetic Resonance Imaging, University Clinical Center of Serbia, 11000 Belgrade, Serbia

**Keywords:** COVID-19, SARS-CoV-2, pulmonary embolism, mechanical ventilation

## Abstract

**Background:** COVID-19 disease caused by severe acute respiratory syndrome coronavirus-2 (SARS-CoV-2); it is characterized by a hypercoagulable state that results in an increased risk for embolic and thrombotic vascular complications. The incidence of pulmonary embolism (PE) in COVID-19 varies between 20 and 30%. In addition to PE, older age, male sex, the presence of comorbidities, invasive mechanical ventilation, and prolonged hospitalization in intensive care units (ICUs) seem to be the main predictors for impaired treatment outcomes in COVID-19. **Materials and methods**: A retrospective observational single-center study was conducted between 1 September 2021 and 24 December 2021 involving 2111 patients admitted to the COVID Hospital “Batajnica”, University Clinical Center of Serbia, Belgrade. A total of 200 consecutive patients were enrolled in study. Patients were divided into two groups—the study group (100 patients), with COVID-19 and PE, and the control group (100 patients), with COVID-19 but without PE. **Results:** According to the multivariate regression analysis, the predictors of impaired outcomes in COVID-19 patients are age (*p* < 0.001; OR 1.134; 95% HR 1.062–1.211), C reactive protein level (CRP) (*p* = 0.043; OR 1.006; 95% 1.000–1.013), invasive mechanical ventilation (IMV) (*p* < 0.001; OR 58.72; 95% HR 13.784–254.189), pulmonary embolism (PE) (*p* = 0.025; OR 3.718; 95% HR 1.183–11.681), and hospitalization in ICU (*p* = 0.012; OR 9.673; 95% 1.660–56.363). **Conclusions:** We report increased mortality and mechanical ventilation rates in COVID-19 patients with acute PE. Older age, elevated levels of CRP, hospitalization in ICU, and PE present independent predictors for impaired outcomes in COVID-19 patients. To determine predictors for treatment outcomes in patients with COVID-19 and their associations with clinical and laboratory parameters

## 1. Introduction

Due to the rapid worldwide spread of severe acute respiratory syndrome coronavirus-2 (SARS-COV-2) infection, several factors relating to impaired treatment outcomes were nominated. It was suggested that older age, male gender, hypertension, diabetes mellitus, obesity, cerebrovascular disease, kidney failure, cancer history, and smoking may be associated with a poor prognosis in SARS-CoV-2-infected patients [1]. The severity of primary respiratory syndromes, including the development of interstitial pneumonia and respiratory failure, was strongly connected with some pre-existing diseases [2]. Namely, patients with chronic obstructive pulmonary disease (COPD) have a 14-fold higher risk factor for poor disease outcomes. Likewise, patients with hypertension have a 2.72–1.44 times greater risk of an aggravating COVID-19 disease than patients without hypertension [3]. The recent COVID-19 pandemic has frequently been associated with numerous micro- and macro-vascular thrombotic events with severe disease outcomes [1,2]. SARS-CoV-2 binds the angiotensin-converting enzyme 2 (ACE 2) receptors on endothelial cells, especially the lungs, kidneys, heart, and liver, leading to endothelial cell damage and a hypercoagulable state [3,4]. COVID-19 implications for coagulation cascades include the overexpression of fibrinogen, thrombin, factor V, VIII, cytokine, and D-dimer. Thrombotic events may occur in both venous and arterial circulation because of extensive inflammation, endothelial dysfunction, platelet activation, and stasis. The influence of SARS-CoV-2 on the cardiovascular system includes pulmonary embolism (PE), myocarditis, acute myocardial infarction, heart failure, and arrhythmias [5]. Pulmonary embolism, as the most commonly found thrombotic vascular complication, presented with a high incidence, i.e., in about 20–30% of hospitalized patients with COVID-19 [6]. In addition to the clinical presentation of COVID-19 pneumonia, the usual symptoms of COVID-19 PE include tachycardia, dyspnea, and hypoxia [7,8]. A previously published paper highlights the important role of D-dimer in the assessment of COVID-19 patients with PE. D-dimer is a factor of the fibrin degradation process and it can be released during the breakdown process of a blood clot, as can be seen in pulmonary embolism [9]. Two types of pulmonary embolism are described—a classic form including deep vein thrombosis and an atypical form with the formation of microthrombus in situ, which is described as a primary pulmonary thrombosis (PPT) [5]. The treatment of confirmed PE in COVID-19 patients according to the guidelines does not differ from those patients without COVID-19 [6,9].

## 2. Materials and Methods

A retrospective observational single-center study was conducted between 1st September 2021 and 24th December 2021, involving 2111 patients admitted to the COVID Hospital “Batajnica”, University Clinical Center of Serbia, Belgrade. A total of 200 consecutive patients were enrolled in the study. Patients were divided into two groups—a study group (100 patients), with COVID-19 and PE, and a control group (100 patients), with COVID-19 but without PE.

The inclusion criteria were as follows: (1) patients presenting a positive result in the reverse transcription-polymerase chain reaction (RT-PCR) assay for SARS-CoV-2 in respiratory specimens (nasopharyngeal swab, tracheal aspirate, bronchial aspirate, or bronchoalveolar lavage fluid) and (2) pulmonary embolism diagnosis made by radiologists on computed tomography pulmoangiography (CTPA). Inclusion criteria for the control group were the same except the fact that patients did not have pulmonary embolism. The exclusion criterion for this study was SARS-CoV-2-negative patients.

### 2.1. Data Extraction

In addition to a positive RT-PCR, data included age, sex, D-dimer, fibrinogen, C reactive protein, interleukin-6, systolic and diastolic blood pressure measured at the time of clinical deterioration, supplemental oxygen administration (nasal catheter, high flow oxygenation, and/or non-invasive positive pressure ventilation), and a history of chronic lung disease, cardiovascular disease, diabetes mellitus, or malignancy. The D-dimer level was measured twice in both study groups—first at the time of hospital admission and second at the time of clinical deterioration including chest pain, hemoptysis, dyspnea, or respiratory failure. Cut off D-dimer values ≥ 0.5 ng/mL had clinical utility (accuracy 70%) in predicting pulmonary embolism. We also extracted data about place of hospitalization and stay in a high-dependency or intensive care unit (ICU). Chest computed tomography (CT) reports were reviewed in order to determine the chest severity score (CTSS), based on the percentage of area involved in each of the five lobes. This can range from 0 (no involvement) to 25 (maximum involvement). CT and CTSS have been devised and validated as a marker of lung involvement in patients affected by SARS-CoV-2. To further ease the diagnosis, the WHO has outlined the typical patterns of lung involvement on CT, which can help to differentiate COVID-19 from other conditions. Also, CTSS may help identify patients with clinical diseases who need early intervention [10].

### 2.2. Imaging Protocol

CTPA was performed in all 200 patients (9.47%) on a 64-section scan (Somatom go.All scanner Siemens Healthineers, Erlangen, Germany) during breath holding, with the injection of a 70–100 mL non-ionic contrast agent (Omnipaque; GE Healthcare, Cork, Ireland) with 100 mL of saline chaster at 4.5/5 mL/s. A small quantity of contrast agent was injected and sequential axial slices at a set region of interest (pulmonary trunk) were acquired to calculate the time of peak contrast enhancement and to determine an optimal scan delay. Images were reconstructed in the axial, coronal, and sagittal plane with 1.25-section thicknesses. Examination findings were reported primarily by chest radiologists. The radiological reports were stored in the RIS platform (Radiology Information System).

### 2.3. Statistical Analysis

The results of this study are presented as numbers (%), means ± standard deviation, or median (25th–75th percentile), depending on the data type and distribution. The groups were compared using parametric (*t*-test) and non-parametric (chi-squared, Fisherʼs exact, and Mann–Whitney *U*) tests. All values of *p* under 0.05 were considered significant. The data were analyzed using SPSS 20.0 (IBM Corp. Released 2011, IBM SPSS Statistics for Windows, Version 20.0, Armonk, NY, USA) and R 3.4.2 software [R Core Team (2017). R:A language and environment for statistical computing. R Foundation for Statistical Computing, Vienna, Austria].

## 3. Results

From 1st September 2021 to 24th December 2021, 2111 patients were admitted to the COVID Hospital “Batajnica”, University Clinical Center of Serbia, Belgrade. The baseline characteristics of 200 consecutive patients are presented in Table 1. ICU admission criteria include patients who required intensive monitoring and oxygen support. The severity of illness scoring system, such as acute physiology and chronic health evaluation (APACHE) and simplified acute physiology score (SAPS), estimate hospital mortality. They cannot be used to predict which patients will benefit from ICU. There were equivalent numbers of patients of both sexes according to presence of PE or age. The average age of patients was 70.58 ± 11.23 years. Patients with acute PE had a lower prevalence of comorbidities, including arterial hypertension (*p* = 0.019), and chronic respiratory disorders, including chronic obstructive pulmonary disease and asthma (*p* = 0.024). Oxygen therapy was much more used in COVID-19 patients with PE (*p* = 0.013). COVID-19 patients with PE were significantly more hospitalized in ICU than patients without PE (*p* < 0.001). A lethal outcome was significantly more common in patients with PE (*p* < 0.001). According to laboratory data, the D-dimer level was significantly higher in PE patients (*p* < 0.001), as well as the CRP (*p* = 0.030) and BNP level (*p* = 0.016), as presented in Table 2. Based on univariate regression analysis, the predictors of impaired outcomes in COVID-19 patients are age (*p* = 0.004; OR: 1.045; 95% HR: 1.014–1.077), diastolic pressure (*p* = 0.027; OR: 0.973; 95% HR: 0.950–0.997), CRP (*p* < 0.001; OR: 1.008; 95% HR 1.004–1.012), IL-6 (*p* = 0.002; OR: 1.003; 95% HR 1.001–1.005), D-dimer level at the time of CTPA (*p* = 0.018; OR: 1.020; 95% CI 1.003–1.0037), BNP level (*p* = 0.001; OR: 1.003; 95% HR 1.001–1.005), CT severity score (*p* < 0.001; OR > 1.124; 95% HR 1.057–1.195), invasive mechanical ventilation (*p* < 0.001; OR: 26/143; 95% HR 11.851–57.672), pulmonary embolism (*p* < 0.001; OR: 3.881; 95% HR 2.038–7.390), and hospitalization in intensive care unit (*p* < 0.001; OR: 52.111; 95% HR 11.851–229.146), as shown in Table 3.

According to multivariate regression analysis, the predictors of impaired outcomes in COVID-19 patients are age (*p* < 0.001; OR 1.134; 95% HR 1.062–1.211), CRP (*p* = 0.043; OR 1.006; 95% 1.000–1.013), invasive mechanical ventilation (*p* < 0.001; OR 58.72; 95% HR 13.784–254.189), pulmonary embolism (*p* = 0.025; OR 3.718; 95% HR 1.183–11.681), and hospitalization in intensive care unit (*p* = 0.012; OR 9.673; 95% 1.660–56.363), as presented in Table 4. The specific characteristics of patients with COVID-19 and PE are presented in Table 5.

## 4. Discussion

Our investigation found worse mortality and morbidity outcomes associated with a diagnosis of acute PE in hospitalized COVID-19 patients. We found that male sex was significantly associated with PE. A recent meta-analysis including more than three million COVID-19 patients found that men were almost three times more likely to require admission to intensive care units and had a 40% higher risk of death than women [10]. In addition to sex differences in the immune system, the male disadvantage in relation to COVID-19 could be explained by androgen influence on endothelial function making the male sex more prone to PE [11]. A multivariable analysis found that age is one of the most important predictors for treatment outcome [12]. Patients with lethal outcomes were significantly older than cured patients. A possible explanation might be that patients with acute PE have a higher prevalence of comorbidities [13]. Contrary to previous studies, our results conclude that comorbidities do not present predictors for acute PE in COVID-19. A possible explanation for this might be related to careful preventive treatment for PE in our patients, as well as the treatment of each comorbid disease. The median levels of arterial pressure in our patients were not pathological, except for low diastolic levels. It was already known that hemodynamic instability in COVID-19 is due to dehydration, sepsis and PE influence of the lower left ventricular charge, and consecutive cardiac shock reflected in low diastolic pressure [4,13]. When it comes to laboratory/monitoring parameters, we noticed higher values of CRP and IL-6. After inflammatory stimulation, vascular endothelial cells, as well as smooth muscles cells, produce large amounts of IL-6. IL-6 intensifies fibrinogen and CRP liver synthesis [14]. Therefore, in cytokine storms with increased levels of IL-6 and CRP due to endothelial dysfunction and hypercoagulability, a higher prothrombotic activity and lower fibrinolytic activity were noticed. Median CRP levels were higher in COVID-19 patients with PE than those without PA (135.5 vs. 95.7 mg/L). Previous findings showed that CRP levels above 108 mg/L were strongly associated with thrombosis (8.3% vs. 3.4%), disease severity (47.6% vs. 9%), and hospital mortality (32.2% vs. 17.8%) [15]. CRP and IL-6 levels seem to be important predictors for treatment outcome in our patients. In patients with clinical deterioration with progressive hypoxemia in cytokine storms, treatment with anti-IL6 agent Tocilizumab proved to be very useful in 8 out of 11 of our patients with PE. Contrary to the levels of CRP and IL-6, the fibrinogen level does not differ between groups. A possible explanation for this might be that equivalent fibrinogen levels in groups with and without PE result from a higher fibrinogen consumption in pulmonary embolism [16]. The D-dimer level was higher in patients with COVID-19 and PE than in those without PE. The D-dimer level measures fibrin degradation products and can be used as a predictor of treatment outcomes in COVID-19 patients [17]. The continued monitoring of D-dimer levels should be especially recommended in the second week of hospitalization. Sudden increases in D-dimer levels mean that COVID-19 patients should undergo CTPA. A univariate analysis of our results showed significantly higher D-dimer levels in the second week of hospitalization of patients with PE. However, cut off D-dimer values for PE in COVID-19 do not exist. Several previous studies have suggested different values between 1.0 and 4.8 mg/L with sensitivity and specificity between 63–100% and 23–84%. Cut off values were at least two times higher in comparison to conventional D-dimer values of 0.5 mg/L, which are usually used in screening for PE in non-COVID-19 conditions [18]. Our results found a similar increase in D-dimer values in both groups before hospitalization, between 1.53 and 2.13 mg/L. Repeated D-dimer values in the second week of hospitalization showed two-fold higher levels among patients with lethal outcomes. It was shown that an increase in D-dimer level of 2.87 two weeks after the beginning of symptoms should be used as a predictive marker for PE with a sensitivity and specificity of 86% and 80% [18,19]. Brain natriuretic protein (BNP) presents an important marker of heart failure. In cases of PE, elevated right ventricular pressure can lead to increased myocardial extension and BNP release. BNP is widely used as an important indicator of 30-days mortality in patients with PE [20]. Our results showed that BNP levels were four times higher compared to reference ranges, and almost five times higher in patients with lethal outcomes, suggesting that BNP levels could be used as a prognostic and predictive biomarker in COVID-19 patients with PE. The CT severity score, previously reported as the percentage of involved pulmonary lobes, was significantly higher in patients who died, but multivariate analysis did not find its importance as a predictor of lethal outcomes in COVID-19 patients with PE. Previous studies have mentioned increased rates of mechanical ventilation, which is consistent with the results of our study [21,22]. Almost all of our patients (96.5%) needed some kind of oxygen therapy during hospitalization, while one-third of them were ventilated with invasive and non-invasive ventilation. We reported a clear association between mortality in PE COVID-19 patients and mechanical ventilation. Pulmonary embolism was usually diagnosed at the time of clinical deterioration in the peri-intubation period. We concluded that mechanical ventilation and hospitalization in ICU were very important predictors of increased mortality in COVID-19 patients with PE. According to CTPA findings, most of our patients (68%) had segmental or sub-segmental PE. Our results were consistent with those of the meta-analysis from Kwee RM et al., suggesting that in cases of sub-massive PE, peripherally located thrombus could play an important role in PE, instead of deep venous thrombosis, as is mostly seen in non-COVID-19 patients [19]. Pulmonary thrombosis in situ is a pathological condition not related to deep venous thrombosis of the lower extremities. The prothrombotic state in COVID-19 patients results from the immuno-thrombotic process characterized by the production of microthrombi in pulmonary capillaries and larger primary thrombi in arterioles. An autopsy of lung samples found pathological evidence for immuno-thrombosis that was not found in other organs such as the heart, kidneys, and brain [23]. Our investigation found that hospitalization in the ICU was five times more common in COVID-19 patients with acute PE. Treatment in the ICU presents a very important and independent predictor of lethal outcome, as was shown before. Acute pulmonary embolism also presents an independent predictor of lethal outcome in hospitalized COVID-19 patients. We have concluded that the risk of lethal outcomes was 2.55 times higher in COVID-19 patients with PE than those without PE.

Our paper highlights the importance of optimal thrombo-prophylactic treatment, the need for CTPA in COVID-19 patients, and D-dimer monitoring, as has been mentioned before [24,25]. Our study has several limitations. First, it is a retrospective single-center study with a relatively small sample size. During the peak of the global pandemic, an enlarged number of patient secondary examinations, including heart ultrasound and lower limb venous Doppler ultrasound, were not performed in the majority of our patients. These factors can result in missing data, particularly in terms of conducting venous ultrasound during the acute phase. Second, data regarding the follow-up period evaluating thrombotic sequelae in patients recovering from COVID-19 are missing. Furthermore, this study reflects real-life patient care with an emphasis on basic clinical data for impaired treatment outcomes in COVID-19 patients.

In conclusion, we report increased mortality and mechanical ventilation rates in COVID-19 patients with acute PE. Older age, elevated CRP level, and hospitalization in ICU present independent predictors for lethal outcomes in COVID-19 patients with PE.

## Figures and Tables

**Table 1 diagnostics-15-01685-t001:** Baseline characteristics of patients.

	Group		*p*
	COVID-19 Without PE (*n* = 100)	COVID-19 with PE (*n* = 100)	
Sex, *n* (%)			
Male	56 (56.0)	62 (62.0)	0.388
Female	44 (44.0)	38 (38.0)	
Age, mean ± sd	70.09 ± 10.80	71.08 ± 11.67	0.534
Number of comorbidities, *n* (%)			
Without comorbidities	7 (7.0)	10 (10.0)	0.836
One	14 (14.0)	14 (14.0)	
Two	28 (28.0)	30 (30.0)	
≥Three	51 (51.0)	46 (46.0)	
Comorbidities, *n* (%)	93 (93%)	90(90%)	
Arterial hypertension	84 (84.0)	70 (70.0)	0.019
Diabetes mellitus	29 (29.0)	26 (26.0)	0.635
Chronic cardiovascular diseases	46 (46.0)	44 (44.0)	0.776
Chronic respiratory disease	16 (16.0)	6 (6.0)	0.024
Malignant disease	9 (9.0)	9 (9.0)	0.158
Oxygen therapy			
Without O_2_	45 (45.0)	29 (29.0)	0.013
≥10 L/min O_2_	32 (32.0)	32 (32.0)	
HFNC/NIV	15 (15.0)	16 (16.0)	
MV	8 (8.0)	23 (23.0)	
Hospitalization, *n* (%)			
High-dependency unit	95 (95.0)	75 (75.0)	<0.001
Intensive care unit	5 (5.0)	25 (25.0)	
Blood pressure (mmHg) *			
Systolic *	132.84 ± 24.30	132.62 ± 20.21	0.945
Diastolic *	75.10 ± 14.04	76.27 ± 11.70	0.523
CT severity score (median values) Treatment outcome	13 (10–18)	15 (11–19)	0.158
Cured	82 (82.0)	54 (54.0)	<0.001
Died	18 (18.0)	46 (46.0)	

Abbreviations: HFNC—high-flow nasal catheter; NIV—non-invasive ventilation; MV—mechanical ventilation; PE—pulmonary embolism; CT severity score—computed tomography score. * Data are median with the interquartile range in parentheses.

**Table 2 diagnostics-15-01685-t002:** Laboratory data.

	Group		*p*
	COVID-19 Without PE (*n* = 100)	COVID-19 with PE (*n* = 100)	
D dimer 1 (ng/mL) *	1.53 (0.87–3.19)	2.13 (0.97–4.70)	0.095
D dimer 2 (ng/mL) *	2.71 (1.32–8.10)	10.60 (3.23–18.94)	<0.001
Fibrinogen (mg/dL) *	4.7 (3.8–6.3)	4.5 (3.2–6.5)	0.307
CRP (mg/dL) *	95.7 (58.3–154.0)	135.5 (75.5–181.5)	0.030
IL-6 (pg/mL) *	29.0 (11.3–55.5)	37.3 (13.0–85.0)	0.154
BNP (pg/mL) *	144 (70–291)	234 (121–602)	0.016

Abbreviations: D-dimer 1—level at the time of hospital admission; D-dimer 2—level at the time of CTPA; CTPA—computed tomography pulmonary arteriography; CRP—C reactive protein; IL-6—interleukin 6; BNP—B-type natriuretic protein level. * Data are median with the interquartile range in parentheses.

**Table 3 diagnostics-15-01685-t003:** Predictors of treatment outcome—univariable logistic regression analysis.

	*p*	OR	95% HR
Age *	0.004	1.045	1.014–1.077
Diastolic pressure *	0.027	0.973	0.950–0.997
CRP (mg/dL) *	<0.001	1.008	1.004–1.012
IL-6 (pg/mL) *	0.002	1.003	1.001–1.005
D -dimer (ng/mL) 2 *	0.018	1.020	1.003–1.037
BNP (pg/mL) *	0.001	1.003	1.001–1.005
CT severity score *	<0.001	1.124	1.057–1.195
MV *	<0.001	26.143	11.851–57.672
PE *	<0.001	3.881	2.038–7.390
ICU *	<0.001	52.111	11.851–229.146

Abbreviations: CRP—C reactive protein; IL-6—interleukin 6; BNP—B-type natriuretic protein level; D-dimer—level at the time of CTPA; CTPA—computed tomography pulmonary arteriography; CT severity score—computed tomography severity score; MV—mechanical ventilation; PE—pulmonary embolism; ICU—intensive care unit; OR—odds ratio; HR—hazard ratio. * Data are median with the interquartile range in parentheses.

**Table 4 diagnostics-15-01685-t004:** Predictors of increased mortality—multivariable logistic regression analysis.

	*p*	OR	95% HR
Age *	<0.001	1.134	1.062–1.211
CRP (mg/dL) *	0.043	1.006	1.000–1.013
MV *	<0.001	58.762	13.784–254.189
PE *	0.025	3.718	1.183–11.681
ICU *	0.012	9.673	1.660–56.363

Abbreviations: CRP—C reactive protein; MV—mechanical ventilation; PE—pulmonary embolism; ICU—intensive care unit; OR—odds ratio; HR—hazard ratio. * Data are median with the interquartile range in parentheses.

**Table 5 diagnostics-15-01685-t005:** Specific characteristics of COVID-19 patients with PE.

CTPA Finding	Number
Massive PE	5 (5%)
Lobar PE	27 (27%)
Segmental PE	61 (61%)
Subsegmental PE	7 (7%)
Antiviral therapy	
Favipiravir	35 (35%)
Tocilizumab	11 (11%)
Remdesivir	54 (54%)
Anti-PE therapy	
Thrombolytic	7 (7%)
LMWH	93 (93%)

Abbreviations: CTPA—computed tomography pulmonary angiography; PE—pulmonary embolism; LMWH—low-molecular-weight heparin.

## Data Availability

Anonymized study data are available from the corresponding author upon reasonable request.

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
