# Peer review of "Predictors of Impaired Treatment Outcomes in COVID-19: A Single-Center Observational Study from Serbia"

_diagnostics, 2025, doi:10.3390/diagnostics15131685_

Round 1
Reviewer 1 Report
Comments and Suggestions for Authors
The article is contribution to expand the experience in treating the patients with COVID 19 especially from higher incidence of pulmonary embolism point of view.
The results of the work are in line with previous research, but it would be of the much greater significance to compare the incidence of pulmonary embolism between the two group of patients : those who were clasiffied, according to the WHO Clinical Progression scale, in the group with moderate diaseses and severe diseases

The article should be revised in relation to the quality of English language.
Author Response
Response to rewiever and commnets
We would like to express our sincere gratitude to the reviewers for taking the time to evaluate our manuscript and for providing their valuable comments and suggestions.
We agree with all the comments and suggestions.
Please find the detailed responses below and the corresponding corrections highlighted in the resubmitted files.
Response 1: We have changed the background with one additional sentence.
Response 2: We have also included exclusion criteria for the study.
Response 3: We had also included inclusion criteria for the control group.
Response 4: We have also included the D-dimer level, measured twice: once at the time of hospital admission and again at the time of clinical deterioration.
Response 5: We have also added the cut-off D-dimer value with clinical utility.
Response 6: In Table 1, we have included the CT severity score for both groups.
We have also added the suggested reference in number 10.
Response 7: In line 100, we have included systolic and diastolic blood pressure measurements taken at the time of clinical deterioration.
Response 8: In line 101, we have included non-invasive positive pressure ventilation.
Response 9: In line 146, we have included the D-dimer level at the time of CTPA.
Response 10: In line 188, we have included low molecular weight heparin.
Response 11: In Table 1, instead of "semi-intensive care unit," we have used "high dependency unit."
Reviewer 2 Report
Comments and Suggestions for Authors
This is a retrospective observational single center study on the predictors for treatment outcome in patients with COVID-19 and its associations with clinical and laboratory parameters. The authors report increased mortality and mechanical ventilation rates in COVID-19 patients with acute PE. Older age, elevated level of CRP, hospitalization in ICU, and PE present independent predictors for impaired outcome in COVID-19 patients. This study may provide some useful information on the treatment outcome and associated parameters in patients with COVID-19. I have some comments.
<Comments>
- Authors demonstrated that predictors of impaired outcome in COVID-19 patients are age, CRP, invasive mechanical ventilation, pulmonary embolism and hospitalization in intensive care unit. In generally, patients who deteriorated vital sign were treated in the ICU. Therefore, it seems natural that ICU admission is a poor prognostic factor. I think that it would be better to indicate ICU admission in more detail, such as SOFA score, APACHE or SAPS3. (ex, average score of SOFA, APACHE or SAPS3 in patients who treated in ICU)
.
- In line 96, Please describe the full term of the abbreviation, “CTPA”.
- In line 104, Please describe the full term of the abbreviation, “CT”.
- In line 109, Correct “CT pulmonary angiography” to “CTPA”.
- In line 104 - 106, authors explained the CT severity score. I think that it would be better to describe more detail.
Author Response
Response to rewiever and commnets
We would like to express our sincere gratitude to the reviewers for taking the time to evaluate our manuscript and for providing their valuable comments and suggestions.
We agree with all the comments and suggestions.
Please find the detailed responses below and the corresponding corrections highlighted in the resubmitted files.
Comment for Reviewer 2:
Response 1.We have added additional sentences regarding ICU admission.
Response 2. In line 96, we have included the full term for the abbreviation "CTPA."
Response 3.In line 104, we have added the full term for the abbreviation "CT."
Response 4. In line 109, we replaced "CT pulmonary angiography" with the abbreviation "CTPA."
Response 5.We have added additional sentences explaining the CT severity score.